# The Role of SUMO E3 Ligases in Signaling Pathway of Cancer Cells

**DOI:** 10.3390/ijms23073639

**Published:** 2022-03-26

**Authors:** Xiaoxia Shi, Yixin Du, Shujing Li, Huijian Wu

**Affiliations:** School of Bioengineering & Province Key Laboratory of Protein Modification and Disease, Dalian University of Technology, Dalian 116024, China; xshi@dlut.edu.cn (X.S.); 22147015@mail.dlut.edu.cn (Y.D.); lsj@dlut.edu.cn (S.L.)

**Keywords:** SUMOylation, SUMO E3 ligases, cancer

## Abstract

Small ubiquitin-like modifier (SUMO)ylation is a reversible post-translational modification that plays a crucial role in numerous aspects of cell physiology, including cell cycle regulation, DNA damage repair, and protein trafficking and turnover, which are of importance for cell homeostasis. Mechanistically, SUMOylation is a sequential multi-enzymatic process where SUMO E3 ligases recruit substrates and accelerate the transfer of SUMO onto targets, modulating their interactions, localization, activity, or stability. Accumulating evidence highlights the critical role of dysregulated SUMO E3 ligases in processes associated with the occurrence and development of cancers. In the present review, we summarize the SUMO E3 ligases, in particular, the novel ones recently identified, and discuss their regulatory roles in cancer pathogenesis.

## 1. Introduction

Protein function is regulated by numerous mechanisms, one of which is post-translational modification (PTM). SUMOylation is a reversible PTM by covalent conjugation of the small ubiquitin-like modifier (SUMO) to target proteins, which is responsible for the target’s localization, activity, and stability in normal and pathological states [1,2,3]. Thus far, five SUMO paralogues have been reported in mammals: SUMO1, SUMO2, SUMO3, SUMO4 [4,5], and SUMO5 [6]. These SUMO paralogues are widely present in eukaryotes and highly conserved across species. There are wide expressions of SUMO1, SUMO2, and SUMO3 in humans, while SUMO4 and SUMO5 are only expressed in certain parts of the body. For example, SUMO4 is only found in the lymph nodes, spleen, kidney, and placenta, and SUMO5 is detected in peripheral blood leukocytes and testes only. The peptide sequence similarity between SUMO2 and SUMO3 is ~97% in humans, and thereby they are often referred to as SUMO2/3 since the antibodies cannot recognize them from each other, and SUMO1 and SUMO2/3 have only 53% sequence similarity [7,8]. SUMO4 has an 86% amino acid homology with SUMO2 [4], and, different to SUMO1 and SUMO2/3, there is a unique proline-90 (Pro-90) residue in SUMO4, which may disrupt the formation of the covalent bond with substrates [9]. SUMO5 is highly homologous to SUMO1; sequence analysis showed that the residues 17–21 IKDED, a conserved SUMO modification motif in SUMO5, are absent in SUMO1 (Figure 1) [6]. 

In all SUMO isomers, the least studied are SUMO4 and SUMO5 till now, their functions are still not well studied. The sizes of these small modifier proteins are around 12KD and through a series of enzymatic cascades, they can covalently bind to substrates (Figure 2). SUMOylation is a dynamical process in which SUMO molecules are conjugated to the lysine residue of a substrate; the attached SUMO isoforms and the physiological state of the cell are important determinants. The SUMO modification consensus motif of most target proteins is “ΨKXE/D”, where Ψ represents a hydrophobic residue, K is the target lysine, X represents any amino acid, and E/D is glutamic acid (Glu) or aspartic acid (Asp) [10,11]. SUMO2/3 carry an internal SUMO consensus motif (SCM) “ΨKXE/D”, which enables the formation of a poly-SUMO chain, while SUMO1 only contains an inverted SUMO consensus site, ExK [12] and often modifies the substrates as a sole molecule, or as the end of a poly-chain of SUMO2/3. Moreover, there is another interaction form between SUMOs and target proteins, which is a non-covalent way through specific SUMO-interacting motifs (SIMs) [13]. The SIMs often consist of four hydrophobic amino acids with the form ΨΨΨΨ, ΨΨxΨ, or ΨxΨΨ, in which Ψ is an amino acid with aliphatic side chain; meanwhile, this side chain is large and non-polar (V, I, or L) and x is usually D, E, S, or T [14,15,16]. Different from the ubiquitination mediated targets degradation (mainly referring to K48-linked ubiquitin chains), SUMOylation predominantly regulates protein localization and activity, and involves in multiple important biologic functions including cell cycle control, gene expression, genome stability, and protein trafficking, which are important for cell and tissue homeostasis. In recent years, many research groups, including our group, have shown that SUMO signaling is implicated in cancers [17,18,19,20,21,22,23,24,25,26]. Many different types of cancers show dysregulation of one or more components of the SUMO machinery, which predominantly results in an increase in SUMOylation.

SUMOylation process contains a highly dynamic enzymatic cascade, which consists of a series of essential components: SUMOs, SUMO activating enzyme E1, SUMO conjugating enzyme E2, SUMO ligases E3, and Sentrin-specific proteases (SENPs). From the identification of SUMO in 1995, only one E1 and one E2 enzyme have been discovered in the SUMO system. E1 is a heterodimeric enzyme, which is composed of the SAE1/SAE2 subunits, and the sole E2 enzyme is Ubc9. In contrast, a growing number of distinct E3 ligases were identified to provide the functional diversity to regulate the activity of many downstream substrate proteins. Mechanistically, the SUMOylation pathway requires several steps: Firstly, SENP converts SUMO into a mature form, which depends on the C-terminal hydrolase activity of the family to cleave SUMO precursors, thereby exposing its diglycine residues. Secondly, mature SUMO is activated by the E1 enzyme, the cysteine residue of E1 is attached by SUMO to form a thioester bond between SUMO and SAE2 in an ATP consuming reaction. Then, SUMO is transferred to the catalytic cysteine of the sole E2 enzyme Ubc9, forming a SUMO-Ubc9 thioester bond via an isopeptide linkage. In the absence of SUMO E3s, very few proteins can be modified by SUMO, such as RanGAP1, and this SUMO modification process can be catalyzed by E3 enzymes by recruiting the target substrate to the Ubc9 charged by SUMO, facilitating Ubc9 to transfer SUMO to the target substrate by altering the structure of the charged Ubc9. SUMO modifications have different forms: mono-SUMOylation, poly-SUMOylation, and multi-SUMOylation [3,27]. Since the SUMOylation process is invertible, SENPs can release SUMO so it can be reutilized. 

SUMOylation, which is distinct from ubiquitination that promotes degradation of its substrate (mainly referring to K48-linked ubiquitin chains), is involved in changing the subcellular localization of proteins, patterns of interaction with other proteins, and often interacts with other post-translational modifications, such as ubiquitination, phosphorylation, acetylation, and methylation [28,29,30]. For example, SUMOylation of promyelocytic leukemia (PML) is required for its localization to PML bodies [31,32]. Studies have shown that the activity of a variety of transcription factors such as Sp3 and p53 can be regulated by SUMOylation [33,34,35,36]. This process usually requires SUMO E3 ligases to increase reaction efficiency and make sure of the specificity. Limited SUMO E3 ligases have been documented over the past 25 years, but more studies have shown that ligases-targeted substrates connect the SUMOylation pathway to diverse biological processes and human diseases, such as neurodegenerative disease, cardiovascular diseases, diabetes, and especially cancers [37,38,39,40]. In this review, the categorization of SUMO E3 ligases and their regulatory roles in cancer pathogenesis are presented.

## 2. The Classification of SUMO E3 Ligases

In the absence of SUMO E3 ligase, SUMO conjugation can also occur, mainly through recognition of the ΨKXD/E motif in the substrate by E2 enzyme Ubc9. Thereby, before the discovery of SUMO E3 ligases, it was thought that the target substrates only needed the E2 enzyme Ubc9 to complete the SUMO process, until the Siz proteins were identified in *S. cerevisiae* functioning as a factor in the same way as E3 ligase during the SUMOylation process [41]. In later research, in humans, dozens of proteins were discovered to function as SUMO E3 ligases. It is SUMO E3 ligases that guarantee the specificity of the substrate and promote the transfer of SUMO from the E2 enzyme to the substrate. SUMO E3 ligases catalyze SUMO transfer mainly by two distinct mechanisms. On the one hand, they can directly interact with the E2–SUMO complex and substrate to facilitate transfer of SUMO to the substrate [15]. On the other hand, they bind the E2–SUMO complex and keep it in a highly efficient orientation for catalysis with E2-mediated substrate specificity in complexes [42,43]. According to their molecular structure and mechanism, we divided the SUMO E3 ligases into five types: the SP-RING domain family; the TRIM superfamily; noncanonical ligases; ligase-like factors; the other SUMO E3 ligases. Table 1 displays a detailed categorization of SUMO E3 ligases.

### 2.1. SP-RING Domain Family

The SP-RING domain stimulates the SUMO ligation and binds with the E2 enzyme Ubc9, which is required for E3 ligase activity in vivo and in vitro [110,111]. In humans, PIAS proteins are the first identified with the SP-RING domain. PIAS proteins are an evolutionarily conserved family of enzymes, which contains six members PIAS1–4 (Table 1). Except for the SP-RING domain, other structural domains in the PIAS proteins also have an effect on the SUMOylation process. In the PIASs’ center, there is a conserved SIM1 domain modulating the interaction between PIAS proteins and SUMO members [112]. Moreover, an extra SIM domain, named SIM2, is also identified in PIAS1–4, while the location and function are a bit different. The SIM2 situates at the C-terminus in PIAS1–3, while in PIAS4 it locates between the SP-RING domain and SIM1. In PIAS1/2, the SIM2 domain is required to bind to SUMO1 and plays a crucial role in the forming of a SUMO1–Ubc9– PIAS1/2 complex, but in PIAS4/y, SIM2 binds to SUMO3 and plays an important role in improving PIAS4 ligase activity [112,113,114].

Initially, PIAS proteins were found to function as inhibitors of STAT transcription factors [115,116,117], but it has become increasingly clear that they regulate a wide range of processes. For example, PIAS proteins regulate DNA damage repair, nuclear trafficking, many nuclear receptors, some transcription factors, and NF-kB signaling [115,118,119]. In addition to their SUMO E3 ligase activity, the PIAS proteins have many other functions. For example, PIASy/4 can activate LEF-1 dependent transcriptional activity even in the absence of the SP-RING domain, which is required for their SUMO E3 ligase activity [120]. 

Human MMS21 (hMMS21) and human zinc finger-containing, Miz1, PIAS-like protein on chromosome 10 (hZimp10) are the other two SUMO E3 ligases containing SP-RING domains [121]. hMMS21 is a subunit of the human SMC5/SMC6 complex; as a SUMO E3 ligase, it promotes the SUMOylation of two substrates, SMC6 and TRAX, which is required for maintaining the genome integrity in response to DNA damage [122]. hZimp10 is detected only in certain human tissues, such as the prostate, ovary, and testis; it is a transcriptional coactivator of p53, Smad3, and the androgen receptor (AR) [123]. While hZimp10 was only confirmed to stimulate the SUMOylation of AR, this SUMOylation enhanced the transcriptional activity of AR in human prostate cancer cells [124]. Since the SP-RING domain in hZimp10 is known to be the binding site for AR, the researchers could design inhibitors with the help of a computer to mimic and occupy the binding site to avoid the SUMOylation of AR, thereby inhibiting the transcription activity of AR. 

### 2.2. TRIM Superfamily

The functionally diverse tripartite motif (TRIM) family is a superfamily with approximately 100 members in the human genome [125]. In this superfamily, TRIM11, TRIM19/PML, TRIM22, TRIM27, TRIM28/KAP1, TRIM32, TRIM33/TIF-1γ, and TRIML2 are also SUMO E3 ligases [126,127,128]. These proteins share a RING domain, one or two B-box domains, and a coiled-coil region. These common domains are located at the N-terminus with the mentioned order, and the more variable regions in TRIM proteins are situated at the C-terminus. Unlike the other SUMO E3 ligases, the RING and B-box domains are both required for TRIM proteins to function. Among them, the RING domain may bind to E2 enzyme Ubc9, and B-box domains are responsible for binding of the substrate, which greatly increases the transfer efficiency of SUMO molecules from Ubc9 to the substrate [126].

TRIM proteins play significant roles in many processes, such as DNA damage repair signaling, tumor suppression, cell growth, etc. [125,129,130]. Among them, TRIM33/TIF-1γ has been reported to act as a SUMO E3 ligase of the transcriptional regulator, SnoN1, regulating the epithelial–mesenchymal transition (EMT) in organoids that derived from human breast cancer cell lines MDA-MB-231 and MCF7, and murine mammary gland epithelia cells [128]. However, different to the other TRIM proteins, the C-terminal PHD and Bromo domains of TRIM33/TIF-1γ are required for their activity as SUMO E3 ligases [128]. The substrates and functional mechanisms of other TRIM proteins as SUMO E3 ligases need to be further clarified.

### 2.3. Noncanonical Ligases (SIM-Containing SUMO E3 Ligases)

Using the SUMO interaction motifs (SIMs), SUMOylated proteins can non-covalently interact with protein partners. SIMs consist of a short stretch of hydrophobic residues flanked by serine residues and/or several acidic residues at the N- or C-terminus [131]. Evidence has shown that some SIM-containing proteins are identified with SUMO E3 ligase activity.

Nucleoporin RanBP2 is a component of nuclear pore complexes (NPCs); it is a large multi-domain protein [132] with essential functions during mitosis and in nucleocytoplasmic transport processes [133,134]. Different to PIAS proteins, there is no SP-RING domain in RanBP2. In RanBP2, there is a fragment of 30-kDa called IR1-M-IR2, which is sufficient for its catalytic activity. It promotes the conjugation of SUMO to the substrate by optimizing the orientation of the SUMO-Ubc9 thioester by combining both SUMO and Ubc9 [42,135,136]. IR1 and IR2 are two short homologous repeats; each IR contains a SIM, which combines the SUMO, specifically for SUMO1 [136]. 

Pc2, also named CBX4, belongs to the family of chromobox (CBX) [137]. The CBX family is a typical component of polycomb group (PcG) complexes that regulate the occurrence and development of cancer by restraining cellular differentiation and self-renewal of cancer stem cells [138]. 

Pc2 recruits Ubc9 and substrates into PcG bodies to function its SUMO E3 ligase activity. The structure–function assay has shown that the E3 ligase activity of Pc2 relies on two domains, the carboxyl-terminal domain for recruiting Ubc9 and substrates and the amino-terminal domain for facilitating the SUMO to the substrate, each of which contains a SIM [139,140].

SLX4, a large multi-domain scaffolding protein, has become a crucial component of multiple pathways, such as genome stability maintenance pathways [141]. Its SUMO E3 ligase activity requires three SIMs and one BTB domain [142]. Similar to other SIM-containing SUMO E3 ligases, such as PanBP2, the SIMs domain in SLX4 helps stabilize the interaction of SLX4 with charged Ubc9-SUMO, and SLX4 preferentially binds poly-SUMO chains. The BTB domain contributes to bind specific substrate, such as XPF, which acts as a substrate adaptor module. The BTB domain, known as POZ (poxvirus and zinc finger) domain, is found at the N-terminus of several C_2_H_2_-type transcription factors and is a protein interaction motif. The expression of a common fragile site could result in mitotic catastrophe, and the SLX4 complex-mediated SUMOylation is crucial to prevent this from happening [142].

ZNF451 protein is a transcriptional co-regulator with zinc finger structure, which partially resides in PML bodies [104]. It consists of two tandem SIMs at the N-terminus, 12 C_2_H_2_ zinc finger domains, and followed by a ubiquitin-interacting motif at the C-terminus. Similar to RanBP2, ZNF451 also interacts with the charged E2 enzyme Ubc9, placing the donor SUMO in an easily released conformation. However, different to RanBP2 specifically binding to SUMO1, the two SIMs of ZNF451 have a preference for combining SUMO2 [143]. 

Human ZNF451 has three isoforms, ZNF451-1, ZNF451-2, and ZNF451-3. They share the common sequence of two SIMs at the N-terminus, which presents the catalytic unit, indicating that the more variable C termini may determine the specificity of substrates. Compared to ZNF451-1, ZNF451-2 lacks amino acids (aa) 870 to 917, and ZNF451-3 is completely different, starting at aa 63, and it has a LAP2α domain at C-terminus. All isoforms share SUMO E4 elongase activity, while only ZNF451-1 was confirmed as a SUMO E3 ligase for the PML [144]. Other isoforms might also function as SUMO E3 ligases, which need to be further studied. 

### 2.4. Ligase-like Factors (Dual Functions as SUMO/Ubiquitin E3 Ligases)

Several ubiquitin ligases also have the activity of SUMO E3 ligases, such as Topors (TOP1-binding arginine/serine-rich), UHRF2, and TRAF7. However, the structure domains that function as a SUMO/ubiquitin E3 ligase are independent. In Topors, the C-terminal sequence is sufficient for a two-hybrid interaction with Ubc9 and SUMO1. Topors-mediated ubiquitination of p53 requires its RING domain at the N-terminus, which is not required for the SUMOylation of p53 [145]. UHRF2 is a RING-type ubiquitin E3 ligase that mediates ubiquitination for cyclin D1 and E1 [146]. Yohan Oh and Kwang Chul Chung showed that UHRF2 also acted as a SUMO E3 ligase for ZNF131 by interacting with SUMO1 and Ubc9 [147]. Its SUMO E3 enzyme activity requires the SRA and NCR domains but not the RING domain. TRAF7 is a member of the TRAF family with the RING and zinc finger domains [148]. It is a ubiquitin E3 ligase, and it can SUMOylate c-Myb at the sites of Lys-499 and Lys-523. TRAF7 binds to the DNA-binding domain of c-Myb via its WD40 repeats, regulating the subcellular localization of c-Myb via SUMOylation [149].

### 2.5. The Other SUMO E3 Ligases

Some proteins were identified with SUMO E3 ligase activity, albeit the essential domains that need to be further clarified. HDAC4 is a member of the histone deacetylase (HDAC) family, its SUMO E3 ligase activity functions to catalyze the SUMOylation of IkBa by interacting with Ubc9 [150], and the interaction of HDAC4 with IkBa needs the C-terminal and N-terminal transcription binding domain (TBD) of HDAC4. Whether the SUMO E3 ligase activity of HDAC4 relies on its TBD domain requires further investigation. Gao et al. showed that HDAC7 could promote the SUMOylation of PML by using its SUMO E3 ligase activity, which played a vital role in the formation of PML nuclear bodies (NBs) [151]. The interaction of HDAC7 with Ubc9 and PML was confirmed in vitro, while the binding domain of HDAC7 as a SUMO E3 ligase has not yet been identified.

SUMOylation is also involved in viral infection by modifying viral proteins or cellular proteins, which are important in antiviral defense [152,153]. Some viral proteins such as adenovirus type 5 (Ad5) E1B-55K [154], Kaposi’s sarcoma-associated herpesvirus (KSHV) K-bZIP [155], and the Ad5 E4-ORF3 [156], were identified as SUMO E3 ligases by an in vitro SUMOylation assay. Ad5 E1B-55K and K-bZIP can both SUMOylate p53, and the Ad5 E4-ORF3 protein has multiple substrates, including TIF-1γ, transcription factor II-I (TFII-I), Nbs1, and Mre11. Thus far, the structural domains of these proteins for SUMO E3 ligase activity have not yet been clarified. 

In summary, more SUMO E3 ligases have been identified, while a lot of work needs to be conducted to investigate the detailed structure–function mechanism of these proteins as SUMO E3 ligases to find whether the new structures are essential for the SUMO E3 ligase activity and whether new substrates can be SUMOylated by current SUMO E3 ligases.

## 3. SUMO E3 Ligases: Dysregulation and Their Role in Human Cancer

With the application of advanced technologies and assay methods, more and more proteins have been identified with functions as SUMO E3 ligases, and numerous functional mechanism studies have been conducted and suggested that SUMO E3s play important roles in many cancers by regulating the protein localization, protein stability, apoptosis, transcriptional regulation, DNA damage repair, and EMT. Much of what is known about the function of SUMO E3 ligases in cancer stems primarily from studies of the PIAS family [38]. In the following sections, the detailed links between SUMO E3 ligases and cancers are described (Figure 3). As well as the PIAS family, we will put our focus more on newly discovered families. 

### 3.1. SUMO E3 Ligases and the Regulation of p53 Signaling Pathway

The tumor suppressor protein p53 plays a critical role in the regulation of the cell cycle, DNA damage, apoptosis, and senescence. Mutations in p53 are frequently found in many cancers [157]. In vitro assays using either cell-free or cell-culture-based assays have shown that SUMO E3 ligases, including all PIAS members [38,158], Topors [145], viral protein E1B-55K and K-bZIP [153], physically interact with p53. Other SUMO E3 ligases, such as TRIML2 [159], MDM2 [34], Pc2 [160], RanBP2 [161], and TRIM25 [162], can also regulate p53 activity either by directly SUMOylating p53 or by SUMOylating p53-associated proteins in distinct tissue environments. Thus far, lysine residue 386 is the only known SUMOylation site of p53. In the absence of K386, the other lysine residues of p53 can also be SUMOylated, while the specific sites need to be further investigated. 

The PIASs are the major family of E3s for p53 SUMOylation; the activity of some PIAS proteins on p53 has been studied, but the others have not yet been studied in detail. Studies have shown that PIAS4-mediated SUMOylation exerted the suppressive effect on p53 transcriptional activity by assisting the nuclear export of p53 [163,164], while PIAS1 can exert activating effects on p53-driven transcription [165]. Oxidative stress can induce the expression of PIAS1, which promotes p53 modification by SUMOylation, thereby stimulating the expression of the pro-apoptotic regulator Bax [33]. The cellular environment, other PTMs, or interacting proteins may affect the action of PIAS proteins on p53, which may explain the different results described above. 

Decorating p53 with different SUMO molecules by the same SUMO E3 ligase could also result in contradictory consequences. Viral protein E1B-55K mediates the conjugation of p53 with SUMO2/3 during adenoviral infection, which requires the binding of E1B-55K to p53 and nuclear localization of the ligase [154], while Pennella et al. found that the SUMO1 modification of p53 by E1B-55K inhibits its transcriptional activity and exports it from the nucleus [166]. It was also reported that viral protein K-bZIP SUMOylated p53. However, it is unclear whether K-bZIP-dependent SUMOylation either activates or represses p53 [155]. 

For the rest of the SUMO E3 ligases, the majority of them promote the transcription activity of p53 by stabilizing the expression of p53 in the nucleus, leading to the inducement of apoptosis. For example, TRIML2-enhanced p53 SUMOylation resulted in down-regulation of growth arrest genes and increased the expression of pro-apoptotic p53 target genes, leading to an increased ability to induce apoptosis [159]. The SUMOylation of p53 can also be regulated by MDM2-ARF, which increased the transcriptional activity of p53, caused cells to undergo senescence, and inhibited the development of PTEN-deficient tumors [34,167]. Upon DNA damage, Pc2/CBX4 can mediate the SUMOylation of hnRNP K to enhance the transcriptional activity of p53 [160], except for RanBP2 and TRIM25, which, combined with androgen-induced G3BP2, increased the SUMOylation of p53, promoting its nuclear export, which facilitated the cellular proliferation and suppressed apoptosis in prostate cancer [161,162].

Meanwhile, the other p53 family member p73 also undergoes SUMOylation mediated by SUMO E3 ligases. PIAS1 SUMOylated tumor suppressor p73, decreased the p73 transcriptional activity on several genes, such as Bax and MDM [168]. PIASγ/4 can also bind and SUMOylate p73, thereby promoting p73 proteasomal degradation [169]. Both ligases mediated the SUMOylation of p73 and resulted in the inhibition of p21 transcription, thereby having a big effect on the regulation of the cell cycle.

Taken together, these studies suggest that the activity and the destiny of p53 and p53-related proteins can be regulated by the SUMO E3 enzyme, thereby promoting or suppressing the p53 activity, which plays important roles in cancer. 

### 3.2. SUMO E3 Ligases and the PI3K/AKT Signaling Pathway

The occurrence of many diseases, including cancer, is related to the abnormality of the PI3K/AKT signaling pathway. AKT is a serine–threonine kinase that regulates many cellular functions, such as cell proliferation, metabolism, differentiation, etc. AKT activation is mainly mediated by phosphorylation, and is also achieved through other PTMs, such as SUMOylation. PIAS1-mediated SUMOylation on AKT K276 activates the activity of AKT kinase, which promotes cell proliferation, survival, and tumor formation [170,171,172]. In contrast, PIASxα inhibits the PI3K/AKT signaling by SUMOylating the PTEN on K266 [173]. PTEN functions as a tumor suppressor in the PI3K/AKT signaling by catalyzing the conversion of PIP3 to PIP2. The conjugation of SUMO1 to K266 in PTEN enhanced PTEN protein stability by reducing its ubiquitination, thereby inhibiting the activity of AKT [173].

Current understanding of SUMO E3 on the PI3K/AKT signaling axis mainly comes from studies on the PIAS family and the results showed that they have diverse influences. In the future, more studies should be conducted to investigate whether any other SUMO E3 ligases are involved in the PI3K/AKT signaling pathway.

### 3.3. SUMO E3 Ligases and PML Protein

The PML protein is a tumor suppressor and an important organizer of PML NBs. It has been reported that when it is overexpressed, cell proliferation is disrupted, resulting in cell cycle arrest, cell senescence, and apoptosis [174,175], while its knockout or reduction in expression correlates with increased tumorigenesis [176].

The role for PIAS1 in SUMO/RNF4-dependent PML degradation has been described in human lung cancer cells [177]. Koidl and co-workers (2016) reported that ZNF451-1 functions as SUMO2/3 specific E3 ligase for PML [144]. This SUMOylation drives PML degradation by RNF4-dependent ubiquitination in the neuroblastoma N2a cell line of mice; an in vivo study should be the next step to investigate whether ZNF451-1 could become a potential target for neuroblastoma.

### 3.4. SUMO E3 Ligases and Genome Stability

During cellular division, the precise DNA replication is essential to maintain genome stability. When these control mechanisms are disrupted, the genome becomes unstable, a key feature of cancer. SUMOylation is one of the regulatory PTMs that play a pivotal role in controlling replication progression, the repair of DNA, and genome stability.

RIF1 plays a key role in inhibiting DNA end resection and promoting nonhomologous end-joining (NHEJ)-mediated DNA double strand break (DSB) repair in the G1 phase of the cell cycle. PIAS4-mediated SUMOylation of RIF1 is essential if RIF1 needs to be modified by ubiquitination or dissociation from DNA damage sites [178]. The deficiency of PIAS4-induced DNA DSBs’ accumulation of ultrafine anaphase bridges (UFBs) in mitotic cells and RIF1 NBs in G1 cells, therefore resulting in genomic instability [178].

The SLX4 is known as a Fanconi anemia protein, which functions as a tumor suppressor and is involved in DNA repair. Guervilly and colleagues discovered that the SLX4 protein can function as a SUMO E3 ligase that SUMO-modifies not only SLX4 itself, but also the XPF subunit of the DNA repair endonuclease ERCC1-XPF. SUMO-dependent functions of SLX4 are detrimental in response to global replication stress but are critical in response to local replication stress [142]. 

### 3.5. SUMO E3 Ligases and Metastasis

Metastasis describes the process through which malignant cells develop the ability to invade tissues beyond their normal boundaries and seed new tumors at secondary sites [179,180]. Metastasis is a complex process that requires a series of cellular changes, such as angiogenesis, EMT, cell migration, and invasion.

When cells lose epithelial characteristics while gaining mesenchymal characteristics, the process is called EMT. A crucial regulator in the EMT is the transforming growth factor-beta (TGF-β). Dadakhujaev and co-workers reported that the SUMO E3 ligase PIAS1 suppresses TGF-β-induced activation of the matrix metalloproteinase MMP2 in human breast cancer cells [181]. Inhibition of PIAS1 in breast cancer cells promotes metastases in mice in vivo. When PIAS1 cooperates with TIF1γ, the collaboration of the two can regulate the SUMOylation of SnoN and inhibit TGF-β-induced EMT, thereby suppressing the growth and invasion of organoids that derived from MDA-MB-231 cells [182,183]. Other members of the PIAS family, PIAS3 and PIAS4 promote the SUMOylation of Smurf2 and Smad3, respectively, which both mediate the suppression of the TGF-β signal response [184,185,186].

Vimentin (VIM) has a role in maintaining cytoskeleton organization and focal adhesion stability; its increased expression is one of the markers of EMT [187]. Li et al. reported that the K439 and K445 residues of VIM could be modified by PIAS1-mediated SUMOylation [40]. After this modification, the solubility of the VIM protein can be increased, and it can promote cell motility and proliferation, which may result in increased cancer cell invasiveness.

The catalytic activities of focal adhesion kinase (FAK) and Src play key roles in promoting protease-associated tumor metastasis and VEGF-related tumor angiogenesis [188]. PIAS1 can SUMOylate FAK at Lys-152, thereby significantly enhancing its auto-phosphorylation at Thr-397, which activates FAK and enhances its ability to recruit many enzymes including Src family kinases. 

The GTPase Rac1 is a SUMOylation substrate of PIAS3; it is very essential for cell migration that can induce cytoskeletal rearrangements [189]. When the expression of PIAS3 is down-regulated, the migration ability of cells is impaired compared with the control group. In PIAS3 down-regulated cells, phosphorylation of p38, a known downstream mediator of Rac1 signaling and a substrate for SUMOylation modifications in MAPK signaling in the study of Uzoma and colleagues, was impaired [190].

In addition to PIASs, Wang et al. recently reported that ZNF451 played a critical role in EMT by the SUMOylation-dependent stabilization of TWIST2 [106]. The CBX4 has been shown to regulate hTERT, the catalytic component of the human telomerase enzyme, mediated transcription of CDH1, and promoted cell migration and invasion in breast cancer [95].

Taken together, SUMO E3 ligases play crucial roles in human cancers by regulating many cellular processes and some hallmarks. Since most of these studies were performed in cancer cell lines, or organoids from cancer cell lines, the direct correlations of those SUMO E3 ligases with clinical tumor samples are missing. To fill in this part, the expression of SUMO E3 ligases in different types of cancer tissues were explored with the Gene Expression Profiling Interactive Analysis (GEPIA) database (http://gepia.cancer-pku.cn/, accessed date is 18 March 2022) [191]. The results clearly showed that expressions of SUMO E3s are different in normal and tumor tissues from different types of cancers, and indeed most SUMO E3s are up-regulated in tumor tissues, such as in cholangiocarcinoma (CHOL) (Appendix A). In addition, for the same SUMO E3 ligase, such as PIAS1, the expression is up-regulated in CHOL, while being down-regulated in uterine corpus endometrial carcinoma (UCEC). The explanation could be that in different cancers, the PIAS1 may modify different substrates, and in the cellular environment, other PTMs may also play a role. This correlation analysis hints that some SUMO E3s are positively correlated with cancer and the mechanism has been studied in vitro, such as CBX4 in breast cancer. While some SUMO E3s were found to be up-regulated or down-regulated in cancers, the mechanisms have not yet been investigated. Thus, combining the in vitro assay and the clinical sample analysis, we believe that whether the pathogenesis of these cancers is dependent on the SUMO E3 ligase activity will be clarified.

## 4. Conclusions and Future Perspectives

It is quite evident from the findings presented above that the relationship of SUMO E3 ligases with human cancers is tight, and multiple mechanisms are involved. Thus, the SUMO E3 ligase may become a potential biomarker for cancer diagnosis and as a drug target for the treatment of cancer. Due to the extremely strong substrate specificity of SUMO E3 ligases, the use of tumor-associated SUMO E3 ligase inhibitors can minimize off-target side effects compared to the use of E1, E2, or SENPs inhibitors, thereby improving the efficiency of cancer treatment, which would be a promising cancer treatment strategy. However, since SUMO E3 ligases usually have many functions, they are not just involved with SUMO E3 ligase activity, as with PIASs, they also function as inhibitors of STAT transcription factors. This makes the inhibition of SUMO E3 ligases complicated and could be one of the reasons that none of the compounds targeting certain SUMO E3 ligases for cancer treatment is in clinical trials yet. However, with the help of great advances in technology, the large-scale, system-wide SUMO proteomics analysis will identify the specific binding site and catalytic site of certain SUMO E3 ligases [192]. By that time, designing inhibitors or agonists with the help of computers to mimic and occupy the binding sites or the catalytic sites, to prevent or enhance the SUMOylation of substrates, will be possible, and the cell-based high-throughput screen could also be used for discovering and verifying the inhibitors or agonists of SUMO E3 ligases [193].

In contrast with the more than 6000 SUMO conjugated proteins identified, only a small number of SUMO E3 ligases have been discovered so far. More and more SUMOylated proteins are being discovered, yet the number of E3 ligases we know is limited; the gap between the two suggests that unknown SUMO E3s are waiting for us to discover. As well as cancer, the SUMOylated proteins are also associated with many other human diseases, such as neurodegenerative disease, cardiac disease, innate immunity in viral infection, etc. Thus, the identification of novel SUMO E3 ligases can not only open new avenues for cancer therapy, but also for the treatment of other related diseases, and it is reasonable to believe that on the basis of a full understanding of the structure of SUMO E3 ligases, with the improvement in screening technology and the corresponding new discovery of the inhibition and activation mechanisms, more success can be achieved against this unexplored potential target. 

## Figures and Tables

**Figure 1 ijms-23-03639-f001:**
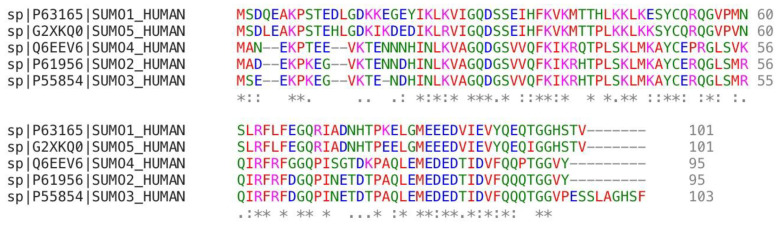
The sequence alignment results of human SUMO isomers. “*” represents the identical amino acid residues; “:” means the roughly similar molecular weight and hydrophilicity; “.” represents there are similar and dissimilar residues in this column.

**Figure 2 ijms-23-03639-f002:**
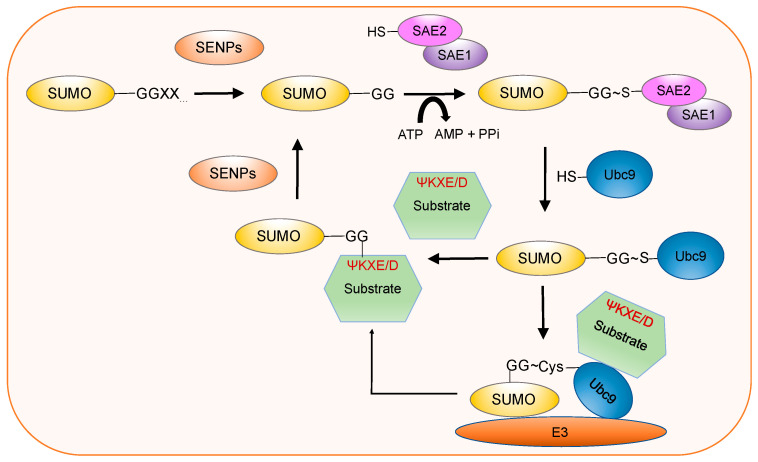
Overview of SUMOylation process. Maturation: Sentrin-specific proteases (SENPs) family converts SUMO into a mature form, which depends on C-terminal hydrolase activity of the family to cleave SUMO precursors, thereby exposing its diglycine residues. Activation: Mature SUMO is activated by the E1 enzyme, the cysteine residue of E1 is attached by SUMO to form a thioester bond between SUMO and SAE2 in an ATP consuming reaction. Conjugation: The activated SUMO is then transferred to the catalytic cysteine of the sole E2 enzyme Ubc9, forming a SUMO-Ubc9 thioester bond via an isopeptide linkage. Ligation: The specific lysine residue of substrate is attached by SUMO with the help of E3 ligases. De-modification: SENPs remove SUMO molecules from substrates, and the released SUMOs can be used in next cycle [15].

**Figure 3 ijms-23-03639-f003:**
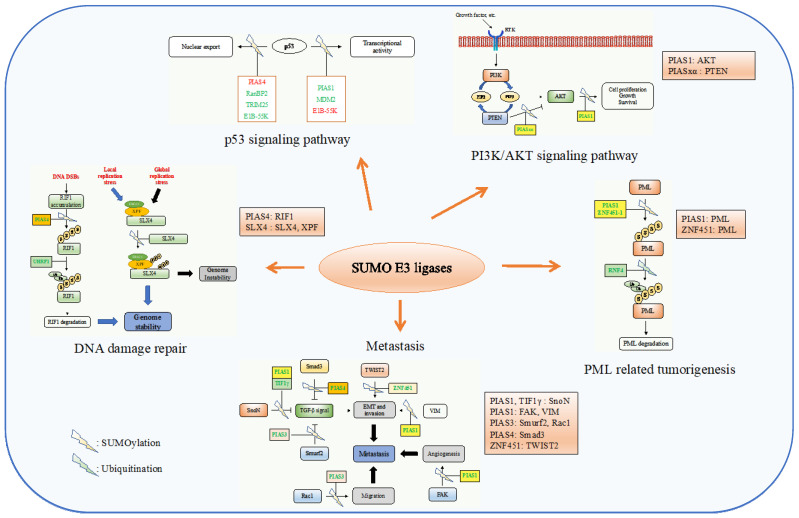
SUMO E3 ligases in the regulation of the hallmarks of cancer. SUMO E3s are involved in cancers by regulating many cellular processes, including the p53 localization, transcription activity, PI3K/AKT signaling, PML protein stability, DNA damage repair, and metastasis. Representative SUMO E3 ligases with their best-defined substrates in these processes are listed. In the p53 signaling pathway, the ligases with green color represent promotion, and those with red color represent inhibition.

**Table 1 ijms-23-03639-t001:** The categorization of SUMO E3 ligases and the cancers they are involved in.

Type	SUMO E3 Ligase	Impacts on Cancers
SP-RING domain family	PIAS1	prostate cancer [44,45], myeloma [38], B-cell lymphomas [38]
	PIASx-α	
	PIASx-β	
	PIAS3	colorectal cancer [38], glioblastoma [46]
	PIAS3β PIAS4	pancreatic cancer [47], colorectal cancer [48], ovarian cancer [49]
	hMMS21	colorectal cancer [50], breast cancer [51]
	hZimp10	
TRIM superfamily	Trim24 (TIF-1α)	hepatic tumors [52], breast cancer [53,54]
	TRIM33	hepatocellular carcinoma [55]
	TRIM28 (KAP1)	hepatocellular carcinoma [55], gastric cancer [56]
	TRIM19 (PML)	acute promyelocytic leukemia [57]
	TRIM27	esophageal squamous cell carcinoma [58], gastric cancer [59], colorectal cancer [60], ovarian cancer [61,62], lung cancer [63,64], renal cancer [65]
	TRIM32	lung cancer [66,67], gastric cancer [68,69], breast cancer [70,71], colorectal cancer [72], pancreatic cancer [73]
	TRIM11	lung cancer [74,75], ovarian cancer [76,77], breast cancer [78,79], gastric cancer [80,81], anaplastic thyroid cancer [82], cervical cancer [83], colon cancer [84], prostate cancer [85]
	TRIM22	gastric cancer [86], endometrial cancer [87], colon cancer [88], lung cancer [89,90]
Noncanonical ligases (SIM-containing SUMO E3 ligases)	RanBP2	colorectal cancer [91], cervical cancer [92], cholangiocarcinoma [93], lung cancer [94]
	Pc2(CBX4)	breast cancer [95], lung cancer [96], cervical cancer [97,98], colon cancer [99], prostate cancer [100], gastric cancer [37]
	SLX4	breast cancer [101], colorectal cancer [102], ovarian cancer [103]
	ZNF451-1/2/3	prostate cancer [104], breast cancer [105,106], hepatocellular carcinoma [106], pancreatic ductal adenocarcinoma [107]
Ligase-like factors (dual functions as SUMO/ubiquitin E3 ligases)	Topors	colorectal cancer [108]
	UHRF2	esophageal cancer [109]
	TRAF7	
The other SUMO E3 ligases	HDAC4	
	HDAC7	
	Viral proteins: Ad5 E1B-55K, K-bZIP, Ad5 E4-ORF3	

Note: the listed SUMO E3 ligases are involved in different cancers, but the SUMOylated substrates in some cancers are unknown.

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
