# Peer review of "The Role of SUMO E3 Ligases in Signaling Pathway of Cancer Cells"

_ijms, 2022, doi:10.3390/ijms23073639_

Round 1
Reviewer 1 Report
X Shi et. al. “The role of SUMO E3 ligases in Cancers”
This literature review presents a description on the different reported SUMOyolation modifications currently known, the mechanisms of SUMOylation and highlights many current investigations on the links of SUMOylation and their associations and reported mechanisms in cancer.
Overall, the manuscript provides a reasonably extensive review of the literature but does not provide any real critical analysis of this literature. Nonetheless, this form of a non-critical literature review/summary merits publication a compendium of reported manuscripts.
While overall the manuscript is well organized and written, there are some point I would recommend the authors to consider.
1) Editorial. While overall the quality of the written document is good there are some minor grammatical corrections that should be made throughout. For example, I will focus on Page 4.
Line 125 - “In human, PIAS proteins are the first ones with SP-RING domain.” Should this be? “In humans, PIAS proteins are the first ones identified with SP-RING domains”?
Line 136 – “…identified as inhibitor of STAT transcription…” should be “…identified as inhibitors of STAT transcription…”
Line 145 – “…SUMO E3 ligases containing SP-RING domain [121] “ should be “…SUMO E3 ligases containing SP-RING domains [121]”
2) SUMO sequences.
With the discussion of the differences in sequences of the different SUMOs and some description of the particular sequences in these SUMO proteins. I would have benefited from a figure listing and aligning these sequences for a visual reference.
3) Misrepresentation of ubiquitination
There are multiple references in the document which repeated state that SUMO is unlike ubiquitination , does not lead to protein degradation.
P2 line 52
P3 line 90
This is somewhat misleading as the authors are actually referring to the ubiquitination with K48 linkages. This is not the case with many with k63 and other linkages where ubuitination has a structural role for trafficking and complex assembly etc. These points should be clarified, otherwise the statements are misleading for a non-specialist reader.
4) Figure 2. Some of the text in Figure 2 is not sufficiently large enough to be read clearly.
5) As presented with respect to cancer, the literature summarized highlights some of the mechanistic insights from these reports. But it would be of interest as well to know which (if any) have results from clinical samples as well.
Reviewer 2 Report
The authors have carried out a review of literature of SUMO PTMs in cellular proteins. However, other than the table which they mention different cancers, no detailed information on how relevant these proteins in these cancers are provided. What are the levels of these ligases in cancers? There are multiple databases; (transcriptomic and proteomic) to survey the levels in tumors. The authors have not used any.
What is the hypothesis here other than sumoylation is implicated in cancer? This review tends to superficially touch about the role of E3 ligase in cell signaling.
The title is misleading as most of the review is about how sumoylation works and different sumo ligases and subsequent signaling.
Major
There is no intuition to how you can hit any of the cancer targets. For Eg;
Line 153: How can you target hZimp10?
Line 167: which cancer would you target using TRIM33/TIF-1γ? What is the model here? Cell culture? Mouse? Tumor?
Line 357mouse neuroblastoma N2a cell- 351 line, which indicates that ZNF451-1 might be a potential target for neuroblastoma. Just because the experiments were done in N2a cell line does not mean that this protein can be targeted.
Since the authors mentioned viral proteins, I would have liked to see more information about sumoylated viral proteins and their implication in tumor development. For eg, HPV, MCPV, EBV etc. but those are reviewed already in htps://doi.org/10.21775/cimb.035.001.
The future perspectives end on a negative note saying there are no inhibitors. How would you develop specific a E3 inhibitor? What are the in vitro assays that one can develop? Any literature? How can specific cancers be targeted?
Minor
Rephrase table 1 legend.
Round 2
Reviewer 2 Report
The manuscript has been improved significantly. I have no further comments.